# Finite element modeling of asymmetric expansion using ATOZ™ expander

Abdelhak Ouldyerou[1,2]*, Peter Ngan[2], Khaled Alsharif[2], Ali Merdji[3], Osama M. Mukdadi[1]*

**1** Department of Mechanical, Materials and Aerospace Engineering, West Virginia University, Morgantown, West Virginia, United States of America, **2** Department of Orthodontics, School of Dentistry, West Virginia University, Morgantown, West Virginia, United States of America, **3** Department of Mechanical Engineering, Faculty of Science and Technology, University of Mascara, Mascara, Algeria

* ouldyerou@gmail.com (AO); Sam.Mukdadi@mail.wvu.edu (OMM)

## Abstract

Asymmetric dentoskeletal expansion is one of the common complications associated with rapid palatal expansion (RPE), often leading to functional and aesthetic concerns. This study investigates the biomechanical performance of ATOZ™ expander using finite element method, with a focus on addressing asymmetrical expansion through expander design modifications. A 3D skull model incorporating teeth and sutures were reconstructed, and two scenarios were analyzed. Model I featured symmetrical suture maturation (stage B), and Model II with asymmetrical suture maturation, particularly more mature zygomatic sutures on the right side (Stage C). Five expander configurations were evaluated in each model. A 0.3 mm total displacement (0.15 mm per side) was applied parallel to the midpalatal suture. Results showed that in Model I, configurations 1, 2, and 3 resulted in relatively symmetrical expansion, while configurations 4 and 5 introduced asymmetry. In Model II, all configurations except configuration 4 demonstrated significant asymmetrical expansion toward the less mature left side. Configuration 4, which included one-sided dental support on the left, was found to redistribute forces and minimize asymmetry. The number of micro-implants had minimal impact on expansion outcomes, while dental support substantially influenced force distribution. These findings suggest that the ATOZ™ expander, especially when configured with one-sided dental support, offers a non-surgical solution to compensate for asymmetrical sutural resistance. This approach may improve clinical outcomes in patients exhibiting unilateral suture maturity and could help reduce the need for surgical intervention such as SARPE.

## 1. Introduction

Micro-implant assisted rapid palatal expander (MARPE) was first described in 2010 by Lee et al. [1] to address the challenges of achieving effective skeletal expansion

**Data availability statement:** All relevant data are within the manuscript and its Supporting Information files.

**Funding:** The author(s) received no specific funding for this work.

**Competing interests:** The authors have declared that no competing interests exist.

in adults and post-pubertal patients. Orthodontic patients in the mixed dentition with immature circummaxillary sutures can be treated with conventional Hyrax rapid palatal expanders (RPE). As maxillary sutures become more mature and interdigitated, expansion of the maxilla with traditional RPE may result in side effects such as alveolar bone bending, tooth tipping, limited expansion, periodontal loss and poor stability. Unfortunately, in adults with a highly interdigitated palatal suture, the only way to achieve absolute skeletal expansion was surgical treatment with LeFort osteotomies. A study in the literature reported that MARPE appliance can produce a similar degree of maxillary bone growth effects as SARPE with a high success rate, ranging from 88.7% to 96.3% in late adolescents and adults [2]. For patients with narrow maxilla and relatively mature sutural development, a new expander called ATOZ™ is commercially available. It was developed in 2017 by Sung-Chul Moon and introduced to the US market in 2021 [3]. This design features 4–5 micro-implants on each side with bases width of 8 mm [4]. The expander is particularly advantageous for patients with craniofacial dysmorphism, as it can address severely narrow palates [3]. Different expanders exhibit subtle variations in their expansion mechanisms and potential side effects. For example, the SARPE procedure can lead to complications such as paresthesia and significant postoperative pain in the immediate aftermath [5], as well as neurosensory deficits over time [6]. Also, the MARPE appliance can cause asymmetric expansion [7,8], dental tipping [9], and gingival inflammation [8]. Although there has yet to be a comprehensive study detailing the complications associated with the ATOZ™ protocol.

In pediatric patients, congenital conditions such as Möbius syndrome or birth trauma can lead to unilateral facial nerve dysfunction and subsequent asymmetrical craniofacial growth, including maxillary underdevelopment. Similarly, trauma to the facial bones during early development can disrupt normal sutural growth patterns, resulting in asymmetrical expansion potential. These underlying factors must be considered in the diagnosis and management of asymmetrical maxillary expansion, as they may influence treatment planning and outcomes [10–12].

Several studies have studied the factors that could contribute to asymmetrical expansion. One of these factors is the separation of the frontomaxillary suture (FMS) [7].

The main objective of this work was to evaluate the asymmetric expansion with five treatment configurations of ATOZ™ expander.

The first null hypothesis posits that asymmetrical expansion is caused by suture maturation. The second null hypothesis posits that with ATOZ™ micro-implants treatment configuration play role in symmetrical expansion.

## 2. Materials and methods

### 2.1. Models

A 3D model of the skull was downloaded from life science database archive [13]. First, skull STL files were imported to the Meshmixer software (version 3.5; Autodesk Meshmixer, San Rafael, Calif). The individual bones of the skull were unified into one single structure. This composite skull was then modified to incorporate various features, including sutures with a thickness of 0.2 mm, achieved through a slicing

method. Midpalatal suture stage B was modeled with scalloped line, as described by Angelieri et al. [14]. Additionally, cancellous bone was created, encased by a 2 mm thick cortical bone layer. A similar approach was applied to the teeth, where the periodontal ligament (PDL) was established by offsetting the root of each tooth by 0.2 mm. **Fig 1** shows the 3D models of skull, teeth, PDL and sutures.

ATOZ™ expander (4th generation) was designed using SolidWorks 2023 (Dassault Système SolidWorks Corp, Waltham, Mass). Two bases and four arms were created with four micro-implants with diameters of 1.6 mm were placed on each side. **Fig 2** shows the basic dimensions of ATOZ™ expander and its orthographic projections.

To assess the asymmetrical skull, different points were taken to measure the distances left and right, as shown in **Fig 3**. **Equation 1** was used to evaluate the asymmetry index. **Table 1** summarizes these measurements.

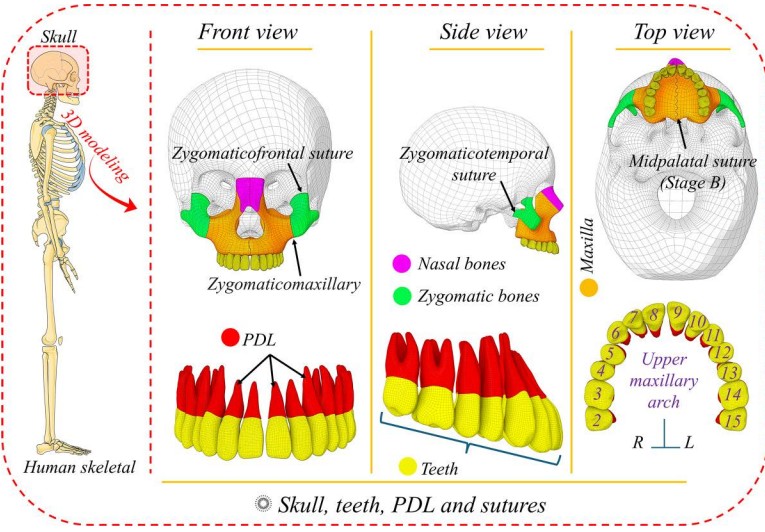

**Fig 1. 3D models of skull, teeth, PDL and sutures.**

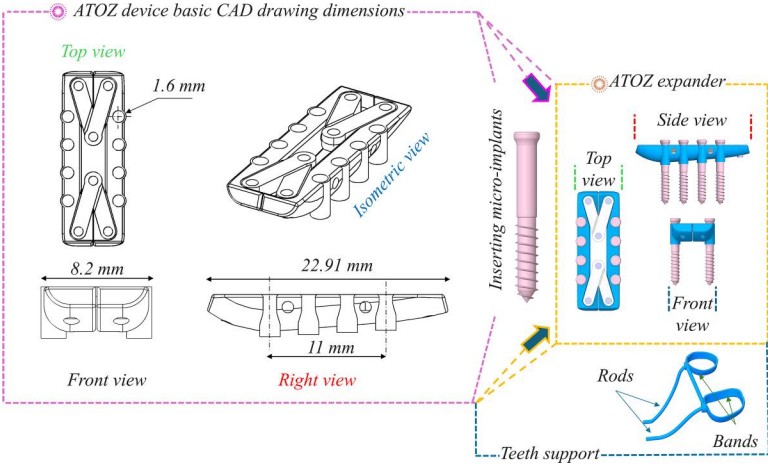

**Fig 2. ATOZ™ expander design.**

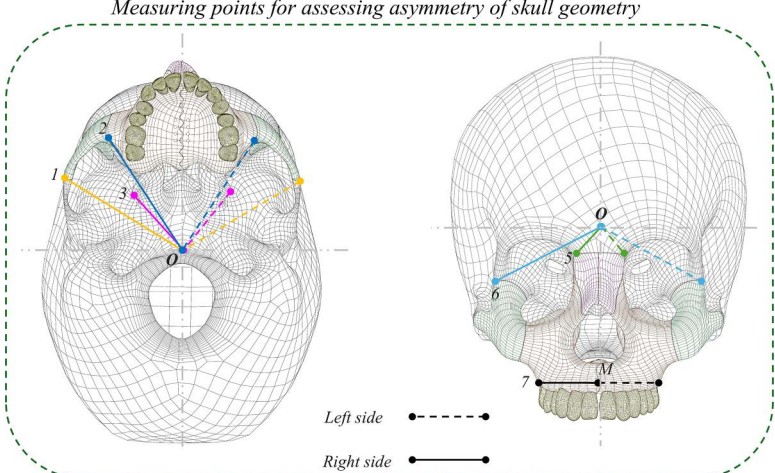

**Fig 3. Measuring points for assessing asymmetry of the skull.**

**Table 1. Measuring distances for assessing asymmetry of the skull.**

|         | Right side | Left side | Asymmetry index |
|---------|-----------|-----------|-----------------|
| O1 (mm) | 73.34     | 73.86     | 0.70            |
| O2 (mm) | 73.47     | 73.65     | 0.24            |
| O3 (mm) | 41.64     | 41.32     | 0.76            |
| O4 (mm) | 105.66    | 105.24    | 0.39            |
| O5 (mm) | 20.68     | 20.73     | 0.24            |
| O6 (mm) | 60.35     | 60.54     | 0.31            |
| M7 (mm) | 30.51     | 30.50     | 0.03            |

$$\text{Asymmetry index} = \frac{\text{Right side - Left side}}{\text{Right side}} \times 100 \tag{1}$$

## 2.2. Three-dimensional finite element analysis

The ATOZ™ expander was integrated into the skull model, and bi-cortical support was established by combining cut micro-implants that penetrated both cortical bone layers. Based on previous reports, the mechanical properties of each component are listed in **Table 2**.

The Young's modulus of the teeth was derived from enamel material properties [17], as the bands are in contact with the crown. It is important to note that the mechanical properties of enamel can vary based on factors such as mineral density, volume fraction, and porosity [18,19]. Stainless steel material was assigned to the expander arms. For its higher biocompatibility, titanium was selected as material for micro-implants and bases.

In this study, two models were investigated based on the resistance of zygomaticofrontal, zygomaticotemporal and zygomaticomaxillary sutures. For model I, all sutures have similar mechanical properties (stage B). While model II, the sutures in the left were more resistant with higher stiffness (stage C) compared to right side. Five treatment configurations were modeled based on the ATOZ™ expander design. Configuration 1 had full micro-implants without any teeth support. Configuration 2 featured two micro-implants on the right side and four on the left side. Configuration 3 included

**Table 2. Material properties assigned to the FE model [15,16].**

| Component | Material | Young's modulus (MPa) | Poisson's ratio |
|---|---|---|---|
| Teeth | Enamel | 83000 | 0.3 |
| Periodontal ligament | Periodontal ligament | 0.068 | 0.45 |
| Micro-implant | Titanium | 114000 | 0.34 |
| Expander device (Arms) | Stainless steel | 210000 | 0.29 |
| Cortical bone | Cortical bone | 13700 | 0.3 |
| Cancellous bone | Cancellous bone | 1370 | 0.3 |
| Suture | Suture (Stage B) | 1 | 0.4 |
|  | Suture (Stage C) | 10 | 0.4 |

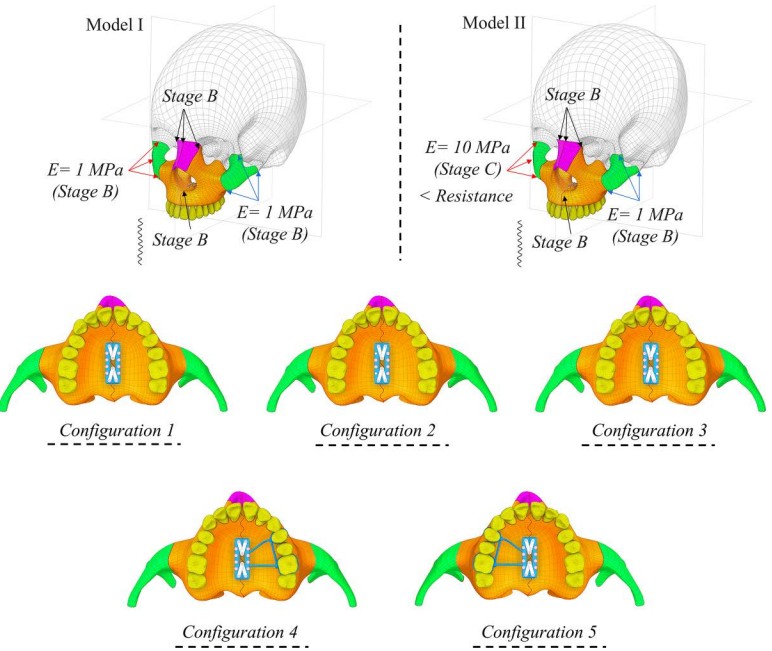

**Fig 4. Finite element setup of symmetric (Model I) and asymmetric (Model II) skull models, and five treatment configurations.**

two micro-implants on the left side and four on the right side. Configuration 4 comprised full micro-implants with one-sided teeth support on the left side, while Configuration 5 had full micro-implants with one-sided teeth support on the right side. **Fig 4** illustrates all these treatment configurations and skull models.

A fixed support was applied at the foramen magnum of the skull to constrain movement in all directions. The expander was activated by applying a sagittal displacement of 0.15 mm on each side at the hinge connecting the arms. For the mesh, finer elements were used for the expander, micro-implants, maxilla, zygomatic and nasal bones. While other regions were meshed with coarse elements. **Fig 5** shows the boundary conditions, applied displacement and mesh.

## 3. Results

To evaluate the expansion movement, total deformation was analyzed based on the obtained results. **Fig 6** illustrates the total deformation distribution of the skull across all models and configurations. The maximum total deformation (shown in red) reached 0.27 mm. The deformation was concentrated in configurations 4 and 5 with model I, as well as in

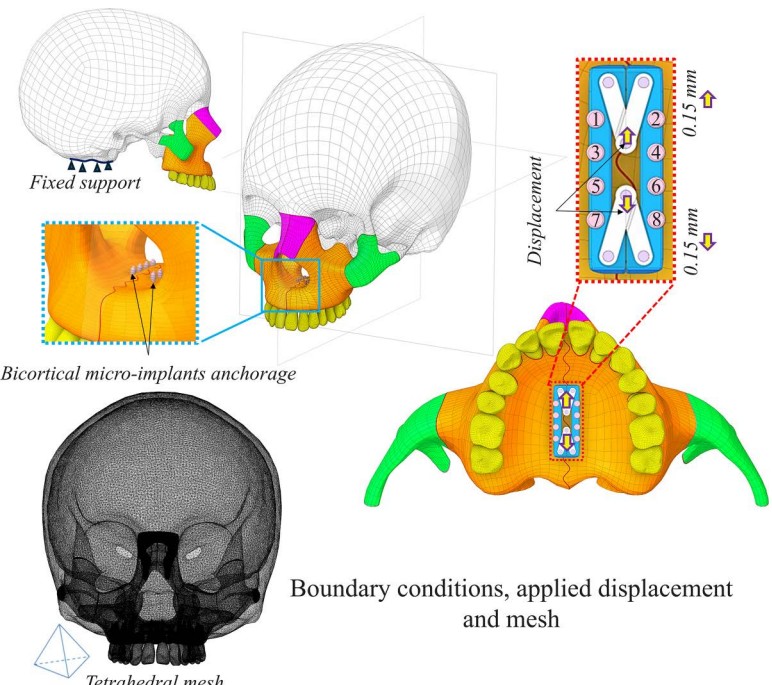

**Fig 5. Boundary conditions, applied displacement (one activation) and mesh.**

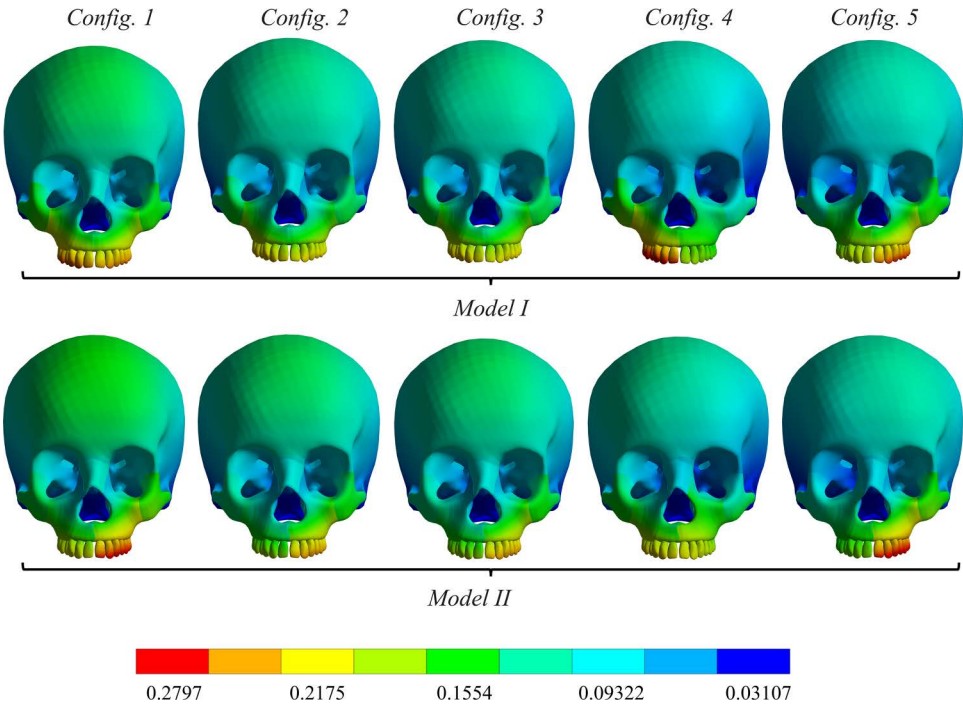

**Fig 6. Total deformation distribution in the skull (in mm).**

configurations 1 and 5 with model II. In model I's configurations 1, 2 and 3, the total deformation was more uniformly distributed compared to configurations 4 and 5 (model I). In contrast, all configurations of model II exhibited an unbalanced total deformation distribution, with greater deformation on the right side than the left, except for configuration 4. Additionally, configurations 2 and 3 displayed comparable outcomes.

It is crucial to evaluate the deformation during the expansion of the zygomatic bone, as these bones significantly influence the overall expansion process of the maxilla. To examine the movement of the zygomatic bone, total deformation was analyzed. **Fig 7** shows the total deformation in left and right zygomatic bone across all configurations and models. Red represents model I while blue represents model II. It was observed that model I exhibited similar deformation in the right and left zygomatic bones across configurations 1, 2, and 3. However, in configuration 4, deformation was greater on the right side, while in configuration 5, it was more pronounced on the left side. Configurations 2 and 3 yielded similar results for both model I and model II. In model II, configuration 4 demonstrated a balanced distribution of deformation. In contrast, configuration 5 of model II exhibited more pronounced shifting compared to model I.

The zygomatic and nasal bones play a vital role in the biomechanics of maxillary expansion, as they absorb and distribute forces generated during the expansion process. Proper stress and strain deformation of these bones are essential for maintaining facial symmetry and preventing adverse effects on occlusion and overall facial function. Therefore, in this study equivalent strain (mm/mm) was evaluated (**Fig 8**) in zygomatic and nasal bones. It was observed that configurations 1, 2 and 3 showed comparable results in strain distribution. Configurations 4 and 5 exhibited more strain distribution in zygomatic bones (for model I), as well as in nasal bones. However, for model II, right zygomatic bone showed more strain distribution than left zygomatic bone. For model I, configuration 4 showed more strain distribution in the right nasal bone than the left, while configuration 5 showed more strain distribution in the left nasal bone.

**Table 3** summarizes the asymmetrical index for distance (M7) to evaluate lateral maxillary expansion. The geometry of our skull model displayed an asymmetrical index of 0.03 before expansion for distance (M7). After expansion, for model I, configurations 1 and 2 maintained the same asymmetrical index of 0.03, while configuration 3 exhibited a slight increase to 0.04. In contrast, configurations 4 and 5 showed more significant increases in the asymmetrical index, with values of 0.22 and 0.10, respectively. For model II, configurations 1, 2, 3, and 5 exhibited increases in the asymmetrical index, with values of 0.12, 0.11, 0.10, and 0.20, respectively. In contrast, configuration 4 displayed a lower asymmetrical index of 0.04.

**Table 4** summarizes von Mises stress in sutures and teeth (in MPa). The von Mises stress values across various sutures and teeth show a diverse range of biomechanical responses under different configurations and models. The midpalatal suture exhibits the highest stress values (1.43 to 1.68 MPa), while the frontonasal suture maintains relatively low stress (0.20 to 0.38 MPa). Other sutures, like the nasomaxillary and zygomaticofrontal, reveal variable stress levels, with the right zygomaticofrontal suture reaching up to 1.66 MPa in some configurations. In terms of teeth, the first molar demonstrates significantly elevated stress values (up to 98.18 MPa), particularly in model I, while the other molars and premolars remain lower, generally ranging between 5 and 9 MPa. Overall, the results indicate that sutural stress varies minimally across configurations, while teeth, especially the first molar, exhibit stress differences between models.

**Fig 9** illustrates the predicted expansion of the maxilla, with movements scaled 35 times. Configurations highlighted in yellow indicated more symmetrical expansion, while those in brown represented more asymmetrical expansion. In model I, configurations 4 and 5 resulted in asymmetrical expansion, whereas model II exhibited asymmetrical expansion across all configurations except for configuration 4.

In **Fig 10**, the angular changes for the first molar across five configurations are compared for both model I and model II. It was observed that in configuration 1 for model I, the right first molar exhibited an angle of 0.26 degrees, while the left first molar measured 0.24 degrees. In configuration 2, the right molar displayed a slight decrease to 0.22 degrees, with the left molar measuring 0.21 degrees. Configuration 3 showed similar values, with the right first molar at 0.22 degrees and the left at 0.21 degrees. However, in configuration 4, the right first molar increased to 0.28 degrees, while the left molar

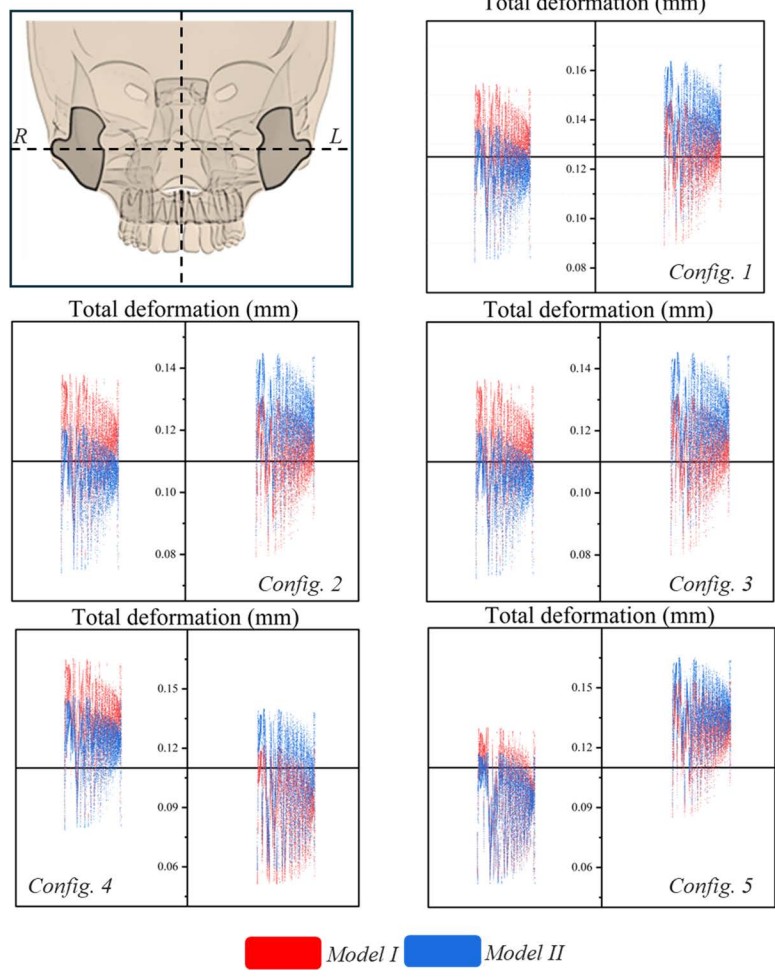

**Fig 7. Total deformation in zygomatic bones (in mm).**

decreased significantly to 0.12 degrees. In configuration 5, the first right molar recorded an angle of 0.13 degrees, whereas the left first molar increased to 0.26 degrees. Overall, model I demonstrated varying angular measurements across configurations, with the right molar generally exhibiting higher angles than the left, particularly in configuration 4.

For model II, it was noted that in configuration 1, the right first molar had an angle of 0.18 degrees, while the left molar displayed a higher value of 0.26 degrees. In configuration 2, both molars experienced a decrease, with the right molar at 0.15 degrees and the left molar at 0.23 degrees. Configuration 3 exhibited a slight increase in the right molar to 0.16 degrees, while the left molar remained at 0.23 degrees. In configuration 4, the right molar rose to 0.20 degrees, but the left molar decreased to 0.13 degrees. Finally, in configuration 5, the right molar recorded its lowest angle at 0.12 degrees, while the left molar increased to 0.27 degrees. Overall, model II exhibited a different trend in angular measurements compared to model I, with the left molar generally presenting higher angles in most configurations, except for configuration 4.

Table 5 summarizes the maximum von Mises stress values for various micro-implants under five configurations, comparing two models (Model I and Model II). In configuration 1, micro-implant 1 experiences the highest stress in model II at 845.67 MPa, slightly exceeding the 828.91 MPa recorded in model I. micro-implant 2 shows comparable results in both models, with model II at 989.36 MPa and model I at 987.75 MPa, indicating consistent performance. micro-implants 3 and

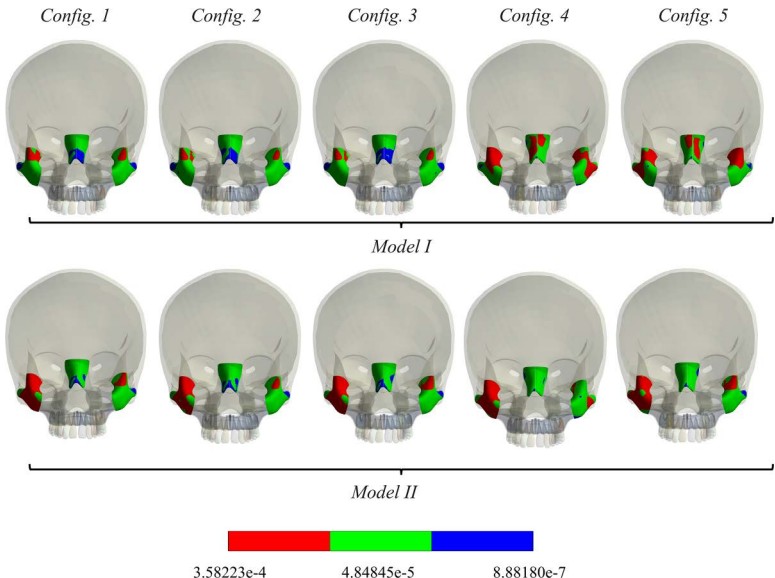

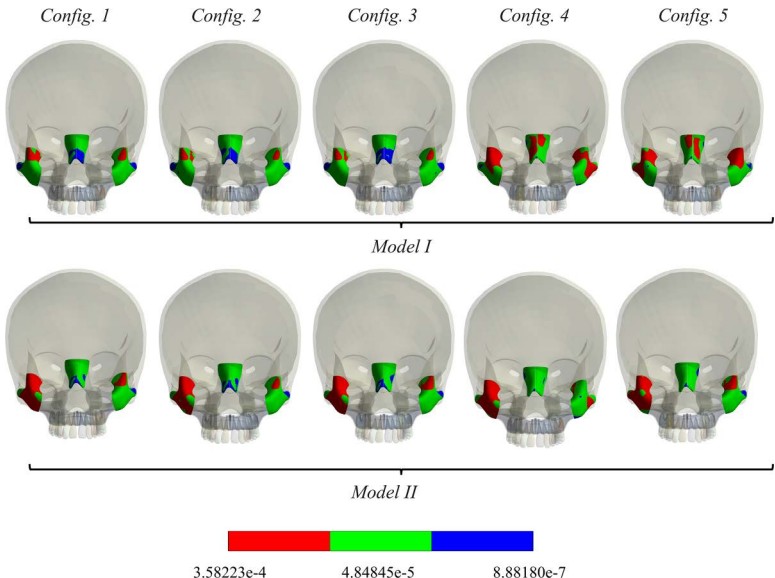

**Fig 8. Equivalent strain in zygomatic and nasal bones (mm/mm).**

**Table 3. The measured asymmetry index for M7 distance.**

|           | Config. 1 | Config. 2 | Config. 3 | Config. 4 | Config. 5 |
| --------- | --------- | --------- | --------- | --------- | --------- |
| **Model I**  | 0.03 | 0.03 | 0.04 | 0.22 | 0.10 |
| **Model II** | 0.12 | 0.11 | 0.10 | 0.04 | 0.20 |

4 exhibit much lower stress levels, micro-implant 3 has values of 245.16 MPa in model I and 211.16 MPa in model II, while micro-implant 4 shows 246.3 MPa in model I and 254.35 MPa in model II, reflecting minor variation.

In configuration 2, micro-implant 2 has distinct values of 895.58 MPa in model I and 887.81 MPa in model II, both higher than the previous configuration. Meanwhile, micro-implants 3 and 4 show increased stress, particularly for micro-implant 4, which approaches 800 MPa in both models. Micro-implant 5 exhibits a significant increase in stress with 874.59 MPa in model I compared to 879.31 MPa in model II.

Moving to configuration 3, micro-implant 1 drops to lower stress levels of 756.91 MPa in model I and 773.75 MPa in model II, while micro-implant 2 remains absent. Micro-implant 3 has relatively low values at 212.27 MPa in model I and 190.05 MPa in model II. In contrast, micro-implant 4 shows substantial stress, with values around 755.61 MPa and 757.7 MPa for models I and II, respectively.

For configuration 4, notable peaks are observed in micro-implants 2 and 7, with micro-implant 2 reaching 1048.3 MPa in model I and 1057.3 MPa in model II, highlighting a significant increase. Micro-implant 7 also exhibits high stress levels, reaching 1023.7 MPa in model I and 1044.7 MPa in model II. Other implants like micro-implant 4 and micro-implant 5 remain relatively stable but show some fluctuations.

In configuration 5, micro-implants generally maintain stress levels consistent with the previous configurations. Micro-implant 1 shows 829.04 MPa in model I and 852.13 MPa in model II, indicating slightly elevated values. Notably, micro-implant 2 retains high stress values at 999.55 MPa and 1000.5 MPa for models I and II, respectively. Overall, stress concentrations were primarily found in areas where the expander contacts the micro-implants. This observation is shown in **Fig 11**, which illustrates von Mises stress in Configuration 4 (model II). In this configuration, von Mises stress values for

**Table 4. von Mises stress in sutures and teeth (MPa).**

|  | Config.1 | | Config.2 | | Config.3 | | Config.4 | | Config.5 | |
|---|---|---|---|---|---|---|---|---|---|---|
|  | Model I | Model II | Model I | Model II | Model I | Model II | Model I | Model II | Model I | Model II |
| Midpalatal suture | 1.68 | 1.65 | 1.47 | 1.57 | 1.47 | 1.43 | 1.64 | 1.60 | 1.63 | 1.61 |
| Frontonasal suture | 0.25 | 0.27 | 0.20 | 0.22 | 0.22 | 0.23 | 0.34 | 0.38 | 0.30 | 0.31 |
| Internasal suture | 0.04 | 0.06 | 0.04 | 0.05 | 0.04 | 0.05 | 0.09 | 0.12 | 0.10 | 0.10 |
| Nasomaxillary suture (L) | 0.18 | 0.21 | 0.16 | 0.19 | 0.14 | 0.16 | 0.32 | 0.38 | 0.19 | 0.20 |
| Nasomaxillary suture (R) | 0.21 | 0.24 | 0.16 | 0.19 | 0.18 | 0.21 | 0.20 | 0.25 | 0.32 | 0.31 |
| Zygomaticofrontal suture (L) | 0.26 | 0.28 | 0.23 | 0.25 | 0.24 | 0.26 | 0.14 | 0.16 | 0.31 | 0.32 |
| Zygomaticofrontal suture (R) | 0.32 | 1.31 | 0.29 | 1.18 | 0.28 | 1.14 | 0.39 | 1.66 | 0.21 | 0.90 |
| Zygomaticotemporal suture (L) | 0.14 | 0.17 | 0.13 | 0.15 | 0.12 | 0.14 | 0.28 | 0.32 | 0.14 | 0.17 |
| Zygomaticotemporal suture (R) | 0.15 | 1.07 | 0.12 | 0.89 | 0.13 | 0.94 | 0.16 | 1.12 | 0.34 | 1.89 |
| Zygomaticomaxillary suture (L) | 0.23 | 0.25 | 0.20 | 0.22 | 0.21 | 0.22 | 0.11 | 0.13 | 0.26 | 0.27 |
| Zygomaticomaxillary suture (R) | 0.30 | 1.37 | 0.26 | 1.19 | 0.26 | 1.20 | 0.33 | 1.50 | 0.18 | 0.83 |
| First molar (L) | 7.02 | 7.48 | 6.09 | 6.49 | 7.40 | 7.82 | 91.86 | 98.18 | 8.18 | 8.42 |
| First molar (R) | 7.48 | 7.38 | 7.64 | 7.51 | 6.54 | 6.43 | 8.30 | 8.21 | 58.18 | 56.18 |
| Second molar (L) | 7.21 | 7.90 | 6.29 | 6.87 | 7.34 | 7.86 | 7.87 | 8.76 | 8.58 | 8.89 |
| Second molar (R) | 6.75 | 6.47 | 6.74 | 6.40 | 5.92 | 5.66 | 7.92 | 7.66 | 7.35 | 7.00 |
| First premolar (L) | 6.00 | 6.21 | 5.23 | 5.39 | 6.22 | 6.39 | 16.20 | 16.79 | 6.82 | 6.91 |
| First premolar (R) | 5.62 | 5.69 | 5.81 | 5.67 | 4.90 | 4.97 | 6.35 | 6.01 | 16.94 | 16.45 |
| Second premolar (L) | 6.00 | 5.47 | 4.61 | 4.75 | 5.71 | 5.89 | 7.78 | 8.35 | 6.12 | 6.22 |
| Second premolar (R) | 5.62 | 5.66 | 6.96 | 6.22 | 5.51 | 4.94 | 7.80 | 6.69 | 9.77 | 8.79 |
| Central incisor (L) | 5.43 | 5.40 | 4.78 | 4.74 | 4.86 | 4.87 | 5.06 | 5.06 | 5.47 | 5.38 |
| Central incisor (R) | 5.14 | 5.19 | 4.59 | 4.44 | 4.56 | 4.59 | 5.17 | 5.02 | 5.17 | 5.00 |
| Lateral incisor (L) | 4.72 | 4.75 | 4.12 | 4.12 | 4.10 | 4.15 | 5.19 | 5.35 | 5.00 | 4.95 |
| Lateral incisor (R) | 4.80 | 4.74 | 4.01 | 3.96 | 4.11 | 4.05 | 5.29 | 5.21 | 5.13 | 4.94 |
| Canine (L) | 4.73 | 4.76 | 4.15 | 4.14 | 4.17 | 4.34 | 5.73 | 6.13 | 5.27 | 5.37 |
| Canine (R) | 4.64 | 4.36 | 4.48 | 4.00 | 3.97 | 3.69 | 5.99 | 5.38 | 5.83 | 5.37 |

L; Left, R; Right.

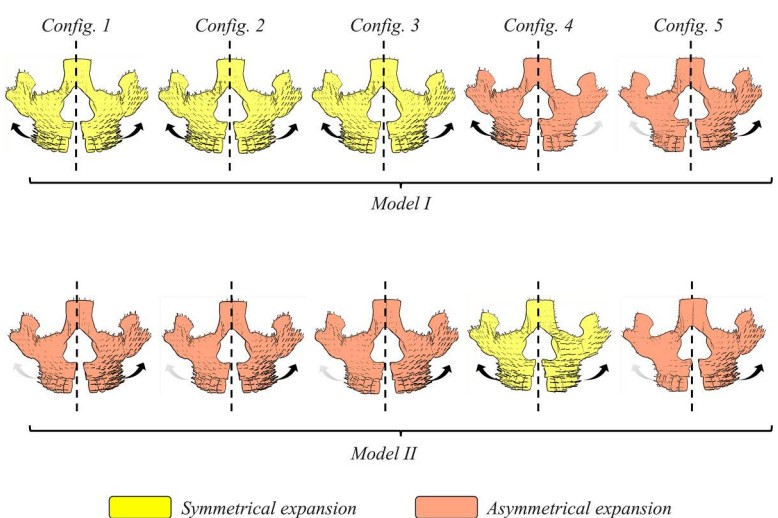

**Fig 9. The expansion prediction (scaled 35 times).**

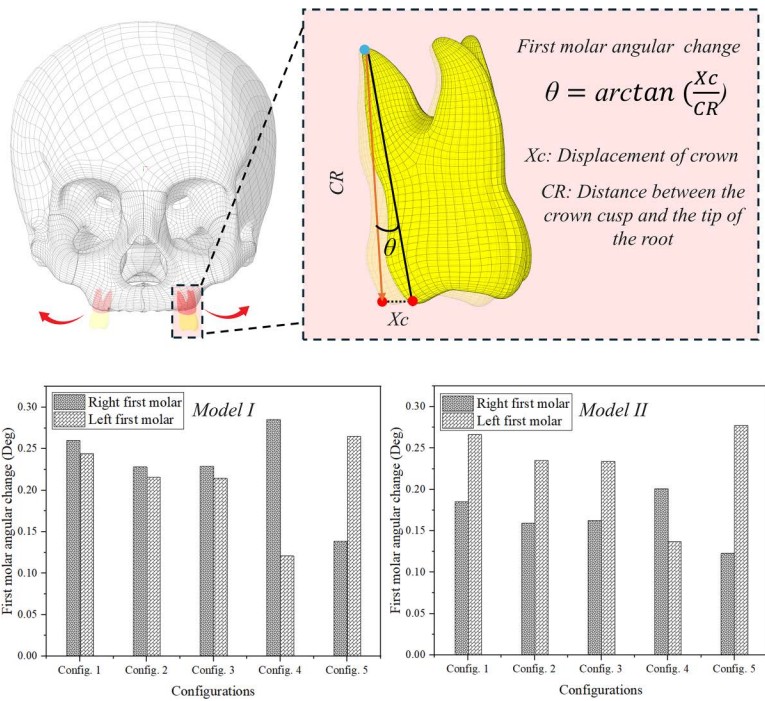

**Fig 10. First molars angular change.**

micro-implants 1, 2, 7, and 8 ranged from approximately 50 MPa to 400 MPa, with some elements exhibiting higher peak values. Meanwhile, micro-implants 3, 4, 5, and 6 showed comparable results, with stress levels range between 25 MPa and 200 MPa. Additionally, it was noted that the teeth support (bands and rods) showed higher stress of 460 MPa.

## 4. Discussion

Orthodontic expanders often lead to unequal expansion between the left and right sides, a discrepancy that can sometimes necessitate surgical correction. Research indicates that variances in surgical cuts during SARPE treatment contribute to this asymmetry, a phenomenon that can be also observed in both bone-borne and tooth-borne expanders [20]. The left-right differences in bilateral structures of the human body can lead to functional or aesthetic issues [21]. This study aims to investigate how sutural changes in the skull can impact the expansion achieved with the ATOZ™ expander with different configurations.

In this study, model I was considered a symmetrical skull, but it is not perfectly symmetrical. This discrepancy could contribute to uneven expansion, in addition to the differences in suture maturation between the right and left zygomatic bones observed in model II. The results showed that maxillary deformation tended to shift more toward the left side in model II, particularly in configurations 1 and 5. This leftward movement is likely due to the softer sutural tissues on the left side.

Regarding the number of micro-implants used, this study found that the number had a minimal impact on expansion. One possible reason for this is the activation rate. In this study, a 0.3 mm activation was applied with the expander, which is within the movement range typically achievable by the micro-implants. Clinically, MSE [9,22–23] procedures often involve just two micro-implants on each side. However, for multiple activations and considering the viscoelastic behavior of bone, increasing the number of micro-implants may play a more significant role in the expansion process. A previous study [24] investigated the ATOZ™ expander with different micro-implant configurations and

Table 5.  Peak von Mises stress in micro-implants (MPa).

| | Config. 1 | | Config. 2 | | Config. 3 | | Config. 4 | | Config. 5 | |
| | Model I | Model II | Model I | Model II | Model I | Model II | Model I | Model II | Model I | Model II |
|---|---|---|---|---|---|---|---|---|---|---|
| Micro-implant 1 | 828.91 | 845.67 | / | / | 756.91 | 773.75 | 838.09 | 862.95 | 829.04 | 852.13 |
| Micro-implant 2 | 987.75 | 989.36 | 895.58 | 887.81 | / | / | 1048.3 | 1057.3 | 999.55 | 1000.5 |
| Micro-implant 3 | 245.16 | 211.16 | 700.6 | 718.67 | 212.27 | 190.05 | 238.58 | 219.32 | 201.92 | 205.66 |
| Micro-implant 4 | 246.3 | 254.35 | 214.99 | 217.99 | 755.61 | 757.7 | 206.63 | 210.15 | 234.38 | 238.97 |
| Micro-implant 5 | 292.86 | 260.6 | 874.59 | 879.31 | 251.37 | 221.51 | 304.53 | 268.92 | 211.4 | 209.44 |
| Micro-implant 6 | 288.67 | 302.88 | 248.35 | 257.25 | 828.78 | 861.56 | 203.86 | 216.93 | 299.53 | 306.33 |
| Micro-implant 7 | 940.68 | 955.08 | / | / | 845.59 | 853.36 | 1023.7 | 1044.7 | 1004.8 | 988.88 |
| Micro-implant 8 | 903.84 | 933.03 | 818.72 | 837.9 | / | / | 971.47 | 1013.8 | 976.63 | 988.92 |

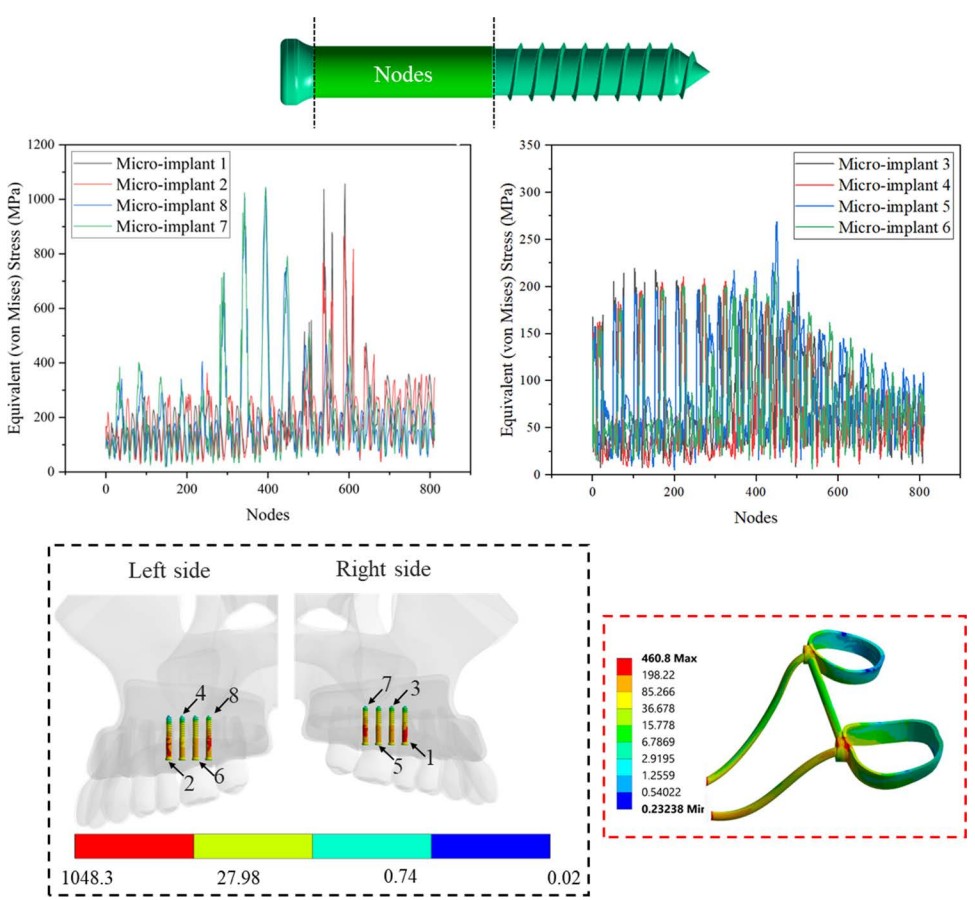

**Fig 11.  von Mises stress (MPa) in micro-implants and teeth support for configuration 4 model II.**

found variations in the initial deformation of the midpalatal suture. However, it reported no significant differences in the stress on the zygomatic and nasal bones or the strain in the periodontal ligaments (PDLs) across the different configurations.

On the other hand, adding dental support to the expander had a significant impact on the expansion process. In model I, configuration 4 showed considerable movement to the left, while configuration 5 demonstrated more movement to the

right. Both configurations resulted in a notable strain on the nasal bones compared to other configurations in model I. This effect was observed when the suture maturation stages on both sides were similar, which was not the case in model II.

In a symmetrical skull scenario (model I), adding dental support to one side (either left or right) can influence the expansion due to the transmission of forces between the two sides. However, when there is an asymmetry in suture maturation between the two sides, as in model II, the outcomes can differ significantly. In configuration 4, where teeth support was added on the left side and the sutures on the right side were more mature, a compensatory balancing of forces occurred, resulting in a more symmetrical expansion. In contrast, configuration 5 demonstrated a less favorable outcome because both the teeth support and the more mature sutures were on the same side (left), leading to more pronounced movement on the left side. This imbalance in force distribution resulted in asymmetric expansion. However, adding dental teeth can affect the teeth surfaces by causing more stress, and dental tipping as results were reported in this regard.

The angular changes in the first molar observed in this study demonstrated that the number of micro-implants had a minimal impact on the expansion when using the ATOZ™ expander after just one turn of activation. However, in contrast, the addition of one-sided dental support led to a lower angular change in model II compared to model I, particularly in configuration 4 within the right first molar. This suggests that while the number of micro-implants may not significantly affect angular movement in the initial activation, the presence of dental support on one side can influence the distribution of forces left and right, resulting in different angular changes between models.

This study is based on several assumptions to simplify the analysis and reduce its complexity. However, these assumptions also introduce certain limitations, which are discussed in the following. The geometry of the ATOZ™ expander was simplified in this study, particularly concerning its components. The jackscrew mechanism was excluded. While this simplification reduces computational complexity, it is important to note that the exclusion of the jackscrew, especially with its threaded design, helps avoid potential computational issues that can arise when applying a displacement. This approach is consistent with other studies in the literature that examine MARPE [25–28] where similar simplifications were made to improve computational efficiency.

In this study, due to a lack of specific data, the behavior of the tissues was assumed to be linear and elastic. However, in reality, bone and sutures exhibit viscoelastic behavior [29,30]. A potential avenue for future improvement would be to incorporate non-linear and viscoelastic properties into the simulation. This would allow for a more accurate representation of tissue behavior, including the ability to account for relaxation time in the bone between activations. Such an enhancement could lead to more precise modeling of the expansion process and better predictions of the long-term effects of treatment.

Despite the limitations discussed, we can conclude that this study provides a promising approach for achieving symmetrical expansion without the need for surgical intervention. The findings suggest that non-invasive methods, such including one sided dental support, may offer an effective alternative for addressing symmetrical maxillary expansion, particularly in cases where surgical options are not preferred. Furthermore, the study highlights the critical role of suture maturation in the expansion process. Recent research has increasingly focused on the degree of suture ossification and its biomechanical resistance, which are key determinants of treatment outcomes. A newly proposed classification system for midpalatal suture maturation, introduced by Chung et al. [31], further contributes to the clinical decision-making process. It builds upon established methodologies, such as those described by Angelieri et al [14], and enhances the clinician's ability to tailor expansion strategies to individual patient needs.

## 5. Conclusion

Based on the ATOZ™ design and the obtained 3D skull model, it was determined that asymmetric expansion during orthodontic treatment may be attributed to differential maturation of the sutures on each side. This asymmetrical expansion was observed in model II, which exhibited a more mature zygomatic suture on the right side compared to the left. Additionally, it was found that the configuration of micro-implants did not significantly impact the expansion, whereas teeth

support played a crucial role. Adding one-sided teeth support could be a viable treatment approach to address asymmetrical expansion, as it helps to compensate for the transmission of strain and stress, allowing for a more evenly distributed expansion.

## Supporting information

**S1 File. Minimal dataset.**
(ZIP)

## Author contributions

**Conceptualization:** Abdelhak Ouldyerou.

**Investigation:** Abdelhak Ouldyerou, Peter Ngan, Khaled Alsharif, Osama M. Mukdadi.

**Methodology:** Abdelhak Ouldyerou.

**Resources:** Peter Ngan, Osama M. Mukdadi.

**Software:** Abdelhak Ouldyerou.

**Supervision:** Peter Ngan, Khaled Alsharif, Osama M. Mukdadi.

**Validation:** Abdelhak Ouldyerou.

**Visualization:** Abdelhak Ouldyerou, Ali Merdji.

**Writing – original draft:** Abdelhak Ouldyerou.

**Writing – review & editing:** Abdelhak Ouldyerou, Peter Ngan, Osama M. Mukdadi.

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
