## [Decision Letter · Decision Letter 0]

7 May 2025

Dear Dr. Mukdadi,

Thank you for submitting your manuscript to PLOS ONE. After careful consideration, we feel that it has merit but does not fully meet PLOS ONE’s publication criteria as it currently stands. Therefore, we invite you to submit a revised version of the manuscript that addresses the points raised during the review process.

We look forward to receiving your revised manuscript.

Kind regards,

Orion Haas Junior, DDS, OMFS, MSc, PhD

Academic Editor

PLOS ONE

**Journal Requirements:**

1. When submitting your revision, we need you to address these additional requirements. Please ensure that your manuscript meets PLOS ONE's style requirements, including those for file naming. The PLOS ONE style templates can be found at https://journals.plos.org/plosone/s/file?id=wjVg/PLOSOne_formatting_sample_main_body.pdf and https://journals.plos.org/plosone/s/file?id=ba62/PLOSOne_formatting_sample_title_authors_affiliations.pdf 2. Please note that PLOS ONE has specific guidelines on code sharing for submissions in which author-generated code underpins the findings in the manuscript. In these cases, we expect all author-generated code to be made available without restrictions upon publication of the work. Please review our guidelines at https://journals.plos.org/plosone/s/materials-and-software-sharing#loc-sharing-code and ensure that your code is shared in a way that follows best practice and facilitates reproducibility and reuse. 3. We note that your Data Availability Statement is currently as follows: All relevant data are within the manuscript and its Supporting Information files. Please confirm at this time whether or not your submission contains all raw data required to replicate the results of your study. Authors must share the “minimal data set” for their submission. PLOS defines the minimal data set to consist of the data required to replicate all study findings reported in the article, as well as related metadata and methods (https://journals.plos.org/plosone/s/data-availability#loc-minimal-data-set-definition). For example, authors should submit the following data: - The values behind the means, standard deviations and other measures reported;- The values used to build graphs;- The points extracted from images for analysis. Authors do not need to submit their entire data set if only a portion of the data was used in the reported study. If your submission does not contain these data, please either upload them as Supporting Information files or deposit them to a stable, public repository and provide us with the relevant URLs, DOIs, or accession numbers. For a list of recommended repositories, please see https://journals.plos.org/plosone/s/recommended-repositories. If there are ethical or legal restrictions on sharing a de-identified data set, please explain them in detail (e.g., data contain potentially sensitive information, data are owned by a third-party organization, etc.) and who has imposed them (e.g., an ethics committee). Please also provide contact information for a data access committee, ethics committee, or other institutional body to which data requests may be sent. If data are owned by a third party, please indicate how others may request data access. 4. PLOS requires an ORCID iD for the corresponding author in Editorial Manager on papers submitted after December 6th, 2016. Please ensure that you have an ORCID iD and that it is validated in Editorial Manager. To do this, go to ‘Update my Information’ (in the upper left-hand corner of the main menu), and click on the Fetch/Validate link next to the ORCID field. This will take you to the ORCID site and allow you to create a new iD or authenticate a pre-existing iD in Editorial Manager.

**Additional Editor Comments:**

The reviewers asked some revision.

You should follow all the recommendations for a future acceptance

Reviewers' comments:

Reviewer's Responses to Questions

**Comments to the Author**

1. Is the manuscript technically sound, and do the data support the conclusions?

Reviewer #1: Yes

Reviewer #2: Yes

2. Has the statistical analysis been performed appropriately and rigorously?

Reviewer #1: Yes

Reviewer #2: Yes

3. Have the authors made all data underlying the findings in their manuscript fully available?

Reviewer #1: Yes

Reviewer #2: Yes

4. Is the manuscript presented in an intelligible fashion and written in standard English?

Reviewer #1: Yes

Reviewer #2: Yes

**Reviewer #1:**  Thanks for your nice work , i appreciate it, i would suggest if you publish another article later with pre and post accompanied with patients photos, and with CBCT imaging to show the effect of ATOZ expander more clearly. By anyway thanks for your hard work.

**Reviewer #2: ** • Both the quality and data presentation of this manuscript are acceptable and of great importance to Maxillofacial surgeon, ENT surgeon, and Orthodontist.

• The manuscript expands our knowledge about the facial expansion and orthodontics.

• The title should be re-write to be more precise & informative.

• The abstract should more informative and should be reflect the content of the article and must be with range of 250 words.

• Four –six keywords should be place not mentioned in title but reflect the manuscript.

• Few more paragraphs should be incorporated to introduction about other studies in diagnosis and management of other causes including congenital and trauma in pediatric age group.

Suggested References:

Aldelaimi TN. New maneuver for fixation of pediatric nasal bone fracture. J Craniofac Surg. 2011;22(4):1476-1478. doi:10.1097/SCS.0b013e31821d1997.

Srivastava D, Singh H, Mishra S, Sharma P, Kapoor P, Chandra L. Facial asymmetry revisited: Part I- diagnosis and treatment planning. J Oral Biol Craniofac Res. 2018;8(1):7-14. doi:10.1016/j.jobcr.2017.04.010

Aldelaimi TN, Khalil AA. Surgical management of pediatric mandibular trauma. J Craniofac Surg. 2013;24(3):785-787. doi:10.1097/SCS.0b013e31828b6c47

• Increase resolution and contrast for each figure within the manuscript with .

• Arrows should be place to each pictures for illustration.

• The author( s) should pay attention to writing and typing errors

• The statements in text are acceptable but more paragraphs about the justification of your findings and comparison with other relevant studies.

• Up to date references should be added to reference list and the old should be omitted.

Good Luck

**Do you want your identity to be public for this peer review?** For information about this choice, including consent withdrawal, please see our Privacy Policy

Reviewer #1: **Yes: ** Ahmed Abouelnour

Reviewer #2: **Yes: ** Tahrir Aldelaimi

---

## [Author Response · Author response to Decision Letter 1]

15 May 2025

Response to Reviewer #1

Comment: Thanks for your nice work , i appreciate it, i would suggest if you publish another article later with pre and post accompanied with patients photos, and with CBCT imaging to show the effect of ATOZ expander more clearly. By anyway thanks for your hard work.

Response:

Thank you very much for your kind words and valuable feedback. We sincerely appreciate your suggestion regarding the inclusion of pre- and post-treatment patient photos along with CBCT imaging to better illustrate the effects of the ATOZ expander. We will take this recommendation into consideration and plan to pursue it in future work.

Thank you once again for your support.

Response to Reviewer #2

Comment 1: Both the quality and data presentation of this manuscript are acceptable and of great importance to Maxillofacial surgeon, ENT surgeon, and Orthodontist.

The manuscript expands our knowledge about the facial expansion and orthodontics.

Response:

Thank you for your positive feedback. We are pleased to hear that you find the quality and data presentation of our manuscript acceptable and valuable to specialists in Maxillofacial Surgery, ENT, and Orthodontics.

Comment 2: The title should be re-write to be more precise & informative.

Response:

Thank you for this suggestion regarding the title of the manuscript.

We agree that a more precise and informative title will help better reflect the scope and focus of the study. Based on your recommendation, the title has been revised.

Finite Element Modeling of Asymmetric Expansion using ATOZ expander

Comment 3: The abstract should more informative and should be reflect the content of the article and must be with range of 250 words.

Response:

Thank you for your comment. The abstract has been revised.

“Asymmetric dentoskeletal expansion is one of the common complications associated with rapid palatal expansion (RPE), often leading to functional and aesthetic concerns. This study investigates the biomechanical performance of ATOZ™ expander using finite element method, with a focus on addressing asymmetrical expansion through expander design modifications. A 3D skull model incorporating teeth and sutures were reconstructed, and two scenarios were analyzed. Model I featured symmetrical suture maturation (stage B), and Model II with asymmetrical suture maturation, particularly more mature zygomatic sutures on the right side (Stage C). Five expander configurations were evaluated in each model. A 0.3 mm total displacement (0.15 mm per side) was applied parallel to the midpalatal suture. Results showed that in Model I, configurations 1, 2, and 3 resulted in relatively symmetrical expansion, while configurations 4 and 5 introduced asymmetry. In Model II, all configurations except configuration 4 demonstrated significant asymmetrical expansion toward the less mature left side. Configuration 4, which included one-sided dental support on the left, was found to redistribute forces and minimize asymmetry. The number of micro-implants had minimal impact on expansion outcomes, while dental support substantially influenced force distribution. These findings suggest that the ATOZ™ expander, especially when configured with one-sided dental support, offers a non-surgical solution to compensate for asymmetrical sutural resistance. This approach may improve clinical outcomes in patients exhibiting unilateral suture maturity and could help reduce the need for surgical intervention such as SARPE.”

Comment 4: Four –six keywords should be place not mentioned in title but reflect the manuscript.

Response:

Thank you for your input. The title has been revised as per reviewer suggestion. Sone of the keywords have been modified.

Keywords: ATOZ Expander, Micro-implants, orthodontic anchorage, Asymmetric Expansion, sutures maturation

Comment 5: Few more paragraphs should be incorporated to introduction about other studies in diagnosis and management of other causes including congenital and trauma in pediatric age group.

Suggested References:

Aldelaimi TN. New maneuver for fixation of pediatric nasal bone fracture. J Craniofac Surg. 2011;22(4):1476-1478. doi:10.1097/SCS.0b013e31821d1997.

Srivastava D, Singh H, Mishra S, Sharma P, Kapoor P, Chandra L. Facial asymmetry revisited: Part I- diagnosis and treatment planning. J Oral Biol Craniofac Res. 2018;8(1):7-14. doi:10.1016/j.jobcr.2017.04.010

Aldelaimi TN, Khalil AA. Surgical management of pediatric mandibular trauma. J Craniofac Surg. 2013;24(3):785-787. doi:10.1097/SCS.0b013e31828b6c47

Response:

Thank you for your suggestion. The suggested references have been added to the introduction.

In pediatric patients, congenital conditions such as Möbius syndrome or birth trauma can lead to unilateral facial nerve dysfunction and subsequent asymmetrical craniofacial growth, including maxillary underdevelopment. Similarly, trauma to the facial bones during early development can disrupt normal sutural growth patterns, resulting in asymmetrical expansion potential. These underlying factors must be considered in the diagnosis and management of asymmetrical maxillary expansion, as they may influence treatment planning and outcomes (10–12).

10. Aldelaimi TN. New Maneuver for Fixation of Pediatric Nasal Bone Fracture. Journal of Craniofacial Surgery. 2011 Jul;22(4):1476–8.

11. Srivastava D, Singh H, Mishra S, Sharma P, Kapoor P, Chandra L. Facial asymmetry revisited: Part I- diagnosis and treatment planning. J Oral Biol Craniofac Res. 2018 Jan;8(1):7–14.

12. Aldelaimi TN, Khalil AA. Surgical Management of Pediatric Mandibular Trauma. Journal of Craniofacial Surgery. 2013 May;24(3):785–7.

Comment 6: Increase resolution and contrast for each figure within the manuscript with .

Response:

Thank you for your comment. All figures are in high resolution (600 DPI). You can click on the link to see the Figure in high format resolution.

Comment 7: Arrows should be place to each pictures for illustration.

Response:

Thank you for your valuable comment. We believe the reviewer is referring to Figure 9. The arrows in this figure represent the direction and magnitude of deformation as generated by ANSYS software. The length and orientation of the arrows are automatically scaled according to the amount of displacement observed in the simulation. For better clarity, we recommend viewing the figure in high resolution.

Comment 8: The author(s) should pay attention to writing and typing errors

Response:

Thank you for your feedback. We have carefully corrected all writing and typographical errors to improve the manuscript.

Comment 9: The statements in text are acceptable but more paragraphs about the justification of your findings and comparison with other relevant studies.

Response:

The authors would like to thank the reviewer for his important comment.

To the best of our knowledge, this is the first in silico study specifically investigating asymmetrical maxillary expansion using the ATOZ™ expander, none have examined the biomechanical behavior of the ATOZ™ design under conditions of differential suture maturation. However, we have added one paragraph and reference to the discussion

A previous study (24) investigated the ATOZ™ expander with different micro-implant configurations and found variations in the initial deformation of the midpalatal suture. However, it reported no significant differences in the stress on the zygomatic and nasal bones or the strain in the periodontal ligaments (PDLs) across the different configurations.

Comment 9: Up to date references should be added to reference list and the old should be omitted.

Response:

Thank you for your suggestion, the references have been updated.

• Half of the cited works were published in the last five years

---

## [Editor Report · Decision Letter 1]

22 May 2025

Finite Element Modeling of Asymmetric Expansion using ATOZTM Expander

PONE-D-25-06726R1

Dear Dr. Ouldyerou,

We’re pleased to inform you that your manuscript has been judged scientifically suitable for publication and will be formally accepted for publication once it meets all outstanding technical requirements.

Kind regards,

Orion Haas Junior, DDS, OMFS, MSc, PhD

Academic Editor

PLOS ONE

---

## [Editor Report · Acceptance letter]

PONE-D-25-06726R1

PLOS ONE

Dear Dr. Ouldyerou,

I'm pleased to inform you that your manuscript has been deemed suitable for publication in PLOS ONE. Congratulations! Your manuscript is now being handed over to our production team.

Kind regards,

on behalf of

Dr. Orion Haas Junior

Academic Editor

PLOS ONE